# Managing Lower Limb Muscle Reinjuries in Athletes: From Risk Factors to Return-to-Play Strategies

**DOI:** 10.3390/jfmk8040155

**Published:** 2023-11-06

**Authors:** Stefano Palermi, Filippo Vittadini, Marco Vecchiato, Alessandro Corsini, Andrea Demeco, Bruno Massa, Carles Pedret, Alberto Dorigo, Mauro Gallo, Giulio Pasta, Gianni Nanni, Alberto Vascellari, Andrea Marchini, Lasse Lempainen, Felice Sirico

**Affiliations:** 1Public Health Department, University of Naples Federico II, 80131 Naples, Italy; 2Venezia FC, 30174 Venezia, Italy; filippo.vittadini@gmail.com; 3Sports and Exercise Medicine Division, Department of Medicine, University of Padova, 35128 Padova, Italy; 4Genoa CFC, 16100 Genoa, Italy; dottor@alessandrocorsini.it; 5Department of Medicine and Surgery, University of Parma, 43126 Parma, Italy; 6Sports Medicine and Imaging Department, Clinica Diagonal, 08950 Barcelona, Spain; info@carlespedret.com; 7Radiology Unit, Casa di Cura Giovanni XXIII, 31050 Monastier, Italy; 8Parma Calcio, 43121 Parma, Italy; 9Isokinetic Medical Group, 40132 Bologna, Italy; 10Policlinico Città di Udine, Viale Venezia 410, 33100 Udine, Italy; info@albertovascellari.it; 11J Medical, 10151 Torino, Italy; andrea.marchini@juventus.com; 12FinnOrthopaedics, Hospital Pihlajalinna, 20520 Turku, Finland; lasse.lempainen@utu.fi

**Keywords:** muscle injury, athletes, rehabilitation

## Abstract

Muscle injuries and subsequent reinjuries significantly impact athletes, especially in football. These injuries lead to time loss, performance impairment, and long-term health concerns. This review aims to provide a comprehensive overview of the current understanding of muscle reinjuries, delving into their epidemiology, risk factors, clinical management, and prevention strategies. Despite advancements in rehabilitation programs and return-to-play criteria, reinjury rates remain alarmingly high. Age and previous muscle injuries are nonmodifiable risk factors contributing to a high reinjury rate. Clinical management, which involves accurate diagnosis, individualized rehabilitation plans, and the establishment of return-to-training and return-to-play criteria, plays a pivotal role during the sports season. Eccentric exercises, optimal loading, and training load monitoring are key elements in preventing reinjuries. The potential of artificial intelligence (AI) in predicting and preventing reinjuries offers a promising avenue, emphasizing the need for a multidisciplinary approach to managing these injuries. While current strategies offer some mitigation, there is a pressing need for innovative solutions, possibly leveraging AI, to reduce the incidence of muscle reinjuries in football players. Future research should focus on this direction, aiming to enhance athletes’ well-being and performance.

## 1. Introduction

Muscle injuries are prevalent in sports, often resulting in significant time loss and setbacks for athletes, especially football players [1]. The economic implications of muscle injuries in professional sports are also noteworthy, with clubs and organizations facing significant financial losses due to player unavailability [2]. Extensive literature has investigated lower limb muscle injuries, injury rates, risk factors, prevention, and rehabilitation programs. Despite significant efforts in rehabilitation programs and return-to-play (RTP) criteria, muscle injury rates have remained high in many sports over the past 20 years, posing a significant challenge for athletes and staff [3].

Muscle reinjury is defined as the recurrence of the same type of injury at the same site within 2 months [4] or up to 1 year [5,6] after an athlete returns to full sports participation. Reinjury can be categorized as early recurrence (within 2 months), late recurrence (between 2 and 12 months), or delayed recurrence (after 12 months) [7]. Understanding the factors contributing to muscle reinjuries is crucial for developing effective prevention and rehabilitation strategies, thereby reducing the impact on athletes’ health.

This article aims to delve into the latest research into lower limb muscle reinjuries in football players, exploring their epidemiology, risk factors, clinical management, and prevention strategies.

## 2. Methods

To conduct a thorough and comprehensive examination of the existing literature on muscle reinjuries, we implemented a rigorous approach in our search and selection process. We utilized various databases, including PubMed, Scopus, and Web of Science, to gather a wide range of articles that are relevant to our topic. To ensure a comprehensive search, we employed a combination of specific keywords and phrases such as “muscle re-injuries”, “prevention strategies”, “clinical management”, “risk factors”, and “previous injuries”.

To determine which articles to include in our review, we established specific criteria for inclusion and exclusion. Articles were considered for inclusion if they were published in peer-reviewed journals, written in English, and primarily focused on topics that align with our interests. On the other hand, articles that were not directly related to the theme of our review or lacked substantial evidence or relevance were excluded from our analysis.

In terms of the analysis parameters, our goal was to provide a comprehensive understanding of muscle reinjuries. We examined the impact of external factors on muscle injuries. Additionally, we explored the significance of previous injuries as a major risk factor. Furthermore, our review emphasized the importance of clinical management and prevention strategies. We also recognized the evolving role of technology in the field of muscle injury management. By considering these various aspects, we aimed to present a holistic view of muscle reinjuries in our review.

## 3. Epidemiology

Recent studies reveal that muscle reinjury rates remain alarmingly high across various sports, underscoring the need for improved prevention and management strategies [8].

The most reinjured muscles are the hamstrings (bicep femoris), rectus femoris, and medial gastrocnemius [5]. Reinjury rates vary between 12% and 43% in different sports, with hamstring musculo-tendinous relapses rating also 50% [9], leading to prolonged out-of-sport periods [10]. Indeed, according to Ekstrand et al., who investigated football players [10], hamstring reinjury, which accounts for about one out of ten of all injuries in field-based team sports, results in a delay in RTP compared with the initial injury. This presents a significant challenge for sports physicians, as 13% of athletes experience hamstring injuries during matches over 9 months [11].

Reinjuries are more common among amateur athletes (48.4%) than among professionals (16.2%) [12,13]. This difference could be attributed to the varying availability of medical staff, diagnostic procedures, appropriate therapies, daily rehabilitation, and RTP scheduling. In particular, RTP is a complex decision involving the athlete, medical staff, and technical staff, who must consider clinical and functional parameters as well as specific aspects of the individual player [14]. Consequently, teamwork, as seen at the professional level, is crucial.

Examining the timing of reinjuries reveals a concerning trend, with many occurring within the first month of an athlete’s return to full participation [15]. The average RTP in muscle injuries often occurs far before the optimal RTP [16]. Muscle healing includes inflammation processes, regeneration processes, and the creation of a healing process in a three-phase transition that requires a specific timeline [17]. On approximately day 21, the scar tissue begins to mold, pulling the myofibrils together and allowing them to bond. From that moment, the myofibrils must further mature until they reach the physiological striated appearance. At 3 weeks, inversion of the collagen III/I ratio also occurs, and the expression of adhesion proteins increases after 2–3 weeks. Therefore, allowing an RTP within 2–3 weeks of the injury forces the player to perform in a timeframe in which the athlete is not biologically ready for a safe return since the scar tissue is still not mature enough. In their systematic review, Van der Horst et al. [18] showed that time from injury is one of the less-used criteria for RTP, and athletes often return to the field earlier than they should. The UEFA Elite Club Injury study by Ekstrand et al. [10] showed the mean of absence days from football for the more common muscle injuries, which is very helpful in guiding the clinical management of athletes (Table 1). Moreover, reinjuries are more frequent in the second half of the season [12], probably because of accumulated fatigue and overload, suggesting a time-dependent mechanism and the tendency to accelerate RTP for crucial matches.

## 4. Risk Factors

Reinjury onset involves various modifiable and nonmodifiable risk factors [19].

Age and previous muscle injury at the same site are the most significant nonmodifiable risk factors supported by scientific evidence [20,21]. Green et al. [22] showed that the age of the player was the strongest predictor (*p* > 0.001) of calf reinjury in their cohort of 149 athletes, and this was recently confirmed by a large systematic review [23]. Figure 1 shows an example of a severe muscle lesion in an old athlete (>40 years old), highlighting how age must always be considered when managing a muscle injury. This imaging serves as a tangible representation of the heightened vulnerability of older athletes to muscle injuries and their potential complications, emphasizing the importance of factoring in age during the diagnosis and management of muscle injuries, given the elevated risk of reinjury in older athletes.

Orchard et al. [24] demonstrated that the absolute risk of sustaining a hamstring strain in a football game was approximately 0.2% for a player with no history of previous history, but 4% for a player with a hamstring strain in the previous 8 weeks. Moreover, in a recent meta-analysis [25], older age (standardized mean difference = 1.6) and recent hamstring strain (risk ratio = 4.8) were considered the main risk factors for a new episode of muscle injury. Athletes who have recovered from a previous muscle injury may exhibit biomechanical abnormalities or neuromuscular control alterations, making them more prone to reinjury, not only in the same structure or muscle.

Recently, an article from the Italian Serie A championship [26] highlighted the potential role of coronavirus disease 19 (COVID-19) in muscle injury, showing how the risk of muscle injury significantly increased after severe acute respiratory syndrome–coronavirus 2 (SARS-CoV-2) infection by 36%. This heightened risk could also conceivably impact the reinjury rates of previously infected players. While the exact pathogenetic mechanism remains elusive, several hypotheses can be postulated. One possibility is that athletes recovering from COVID-19 may experience residual fatigue or diminished physiological resilience, rendering them more susceptible to injuries. The systemic inflammatory response triggered by the virus might also compromise muscle tissue integrity or function. Given the potential severity of the inflammatory process associated with COVID-19, ensuring an adequate recovery period for athletes post-infection is imperative to mitigate injury risks.

Other nonmodifiable risk factors with lower-level evidence include reduced muscle strength [27], low muscle flexibility [28], muscle fatigue [29], and modified characteristics of the muscle after the first injury (such as weak scar tissue, biomechanical abnormalities, or neuromuscular control alterations) [30]. Furthermore, Malliaropoulos et al. [31], in their cohort study about track and field athletes, highlighted how low-grade hamstring muscle lesions had a higher risk of reinjury than high-grade hamstring muscle lesions, underlying how underestimating an injury in terms of rehabilitation care could have serious consequences on the career of an athlete.

The Italian Society of MUscles, Ligaments and Tendons (ISMULT) summarizes epidemiology and risk factors for muscle reinjuries (Table 2) in their latest guidelines [32].

## 5. Clinical Management

Symptoms, diagnosis, and therapy of muscle reinjury are the same as those of the first episode (Figure 2) [33]. The cooperation between coaches, strength and conditioning specialists, physiotherapists, and medical staff is of paramount importance in the rehabilitation of a muscle lesion. Indeed, Ghrairi et al. [34], in their retrospective review of 15 seasons of a professional football team in Dubai, showed a significant increase in the mean number of total injuries, mean number of indirect muscle injuries, and indirect muscle reinjuries during seasons with a poor perceived level of cooperation between these figures.

The muscle tissue repair process is completed in a period that depends on the severity of the lesion. During this period, different well-defined biological phases are involved. Each of these phases must be characterized by a definite type of muscular contraction consistent with the biological condition observed within the injured area. Although rehabilitation is subdivided into a defined number of steps, the duration of each step is different, and progression is not time-based but is based on clinical, functional, and imaging criteria [33]. Therefore, the duration of each phase is consistent with the dynamics of the healing processes occurring in the muscle tissue and with the severity of the injury (Figure 2).

Clinicians often rely on clinical measures such as pain on palpation, muscle strength assessment, and functional tests to guide the rehabilitation process and monitor progress [35]. Whiteley et al. [36] showed that the length of pain on palpation, strength measured in the outer range position, hip flexion active knee extension test, and asking about pain during daily activities are the most useful clinical measures to guide a rehabilitation process in the management of hamstring injury. A reduction in these measures by approximately 50% indicates the completion of half of the rehabilitation process. However, it is crucial to acknowledge the inherent subjectivity in these clinical evaluations. The clinician’s interpretation and the patient’s self-reporting can introduce potential biases, possibly affecting the accuracy and consistency of these measures. Furthermore, a significant challenge arises when players, despite showing no symptoms and having normal examination results and exercise progression, suffer reinjuries. This suggests that even if clinical measures seem normalized, the underlying tissue might not have achieved the necessary quality and maturation, emphasizing the need for more objective and comprehensive assessment tools in the rehabilitation process.

In the realm of sports medicine, a plethora of therapeutic options have been explored to address muscle injuries. Physical therapies, encompassing modalities like cryotherapy, electrostimulation, ultrasound, and manual therapies, are staples in musculoskeletal rehabilitation. However, despite their widespread use, robust scientific evidence supporting their definitive effectiveness remains elusive [33]. This lack of concrete evidence makes the formulation of a universally effective therapeutic strategy for muscle injuries a challenging endeavor. Orthobiologic treatments, which harness the body’s natural healing mechanisms, have gained traction in recent years. Among these, platelet-rich plasma (PRP) injections stand out. PRP involves concentrating platelets from the patient’s blood and injecting them into the injured site, aiming to accelerate tissue repair and regeneration. Preliminary studies and anecdotal evidence suggest that PRP might enhance recovery speed and improve tissue quality post-injury. However, rigorous scientific studies providing strong evidence for its efficacy are still limited [37,38].

Establishing specific clinical and functional return-to-training (RTT) and RTP criteria has been a recent development. In a recent survey on the English Premier League [39], the most used criteria were the absence of pain during muscle palpation, the absence of pain during muscle maximum contraction, the complete recovery of muscle strength and flexibility, and sport-specific functional tests. Furthermore, the approach described for hamstrings could be quite easily used for the rectus femoris and adductors; however, in the case of deeper muscles or complicated injuries involving the muscle–tendon junction or fascia, a more cautious approach is required [40]. The RTT process, therefore, should be as individualized as possible to allow a safe and fast return after a muscle injury, considering some key clinical points: this means controlling all the individualized risk factors. RTP, on the other hand, should be composed of specific assessments, laboratory tests, and field tests tailored for each muscle group [33,41].

## 6. Imaging

Despite the high frequency of muscle injuries in elite athletes and the prime concern being minimizing the number of days lost from sporting activities, there is still a lack of uniformity in the description, diagnostic approach, and grading of muscle injuries [42]. Ultrasound imaging is frequently used in the evaluation of musculoskeletal pathologies as a first-line approach [43], given its wide availability, good tolerability, easy use, fasting, and low cost, if compared with MRI. Moreover, US imaging offers dynamic evaluations in real time, being able to take advantage of the patient’s collaboration to better characterize the lesion [43] and is particularly useful in the serial evaluation of an athlete after a muscle injury. The ultrasound features of muscle strain found in different grades of injury were previously described by Peetrons [44] (Table 3).

In low-grade muscle injury, the reparative process appears as an increase in the echogenicity of the lesion area, with a progressive reduction in its extension [45]. Higher-grade lesions are characterized by the formation of a hematoma. During the reparative process, hematoma undergoes liquefaction resulting in hypoechoic, with progressive resorption and reduction in its extension. Lesion margins will be hyperechoic and echogenic material inside the lesion, representing the deposition of scar tissue, will be observed [44]. Therefore, the role of echography lies in three main aims [44]: assessment of the extent of injury and measurement of the separation between the normal margins; to determine the stage of healing by demonstrating the filling of the hemorrhagic cavity by a hyperechoic tissue corresponding to the healing process; and the assessment of the magnitude of the scar formation. 

Differently, MRI is often the first diagnostic choice in professional athletes because of its accuracy in identifying the site of the lesion and quantifying the percentage of muscle cross-sectional area in the images obtained at the level of maximal abnormality, which is related to grading [46]. A stepwise systematic approach for MRI assessment of muscle injuries has been suggested by Isern-Kebschull et al. [42] (Table 4).

Knowledge of some anatomical characteristics of the distribution of connective tissue and the orientation of fascicles/fibers in the hamstrings, rectus femoris, adductors, and calf muscles is crucial for accurately interpreting MRI findings in the diagnosis of muscle injuries of the lower limbs (Table 5). The distal myotendinous junction (DMTJ) of the biceps femoris has a complex multicomponent anatomy that originates from two zippers (superficial and deep) whose location is important, given that it has been shown to have a different prognosis [49]. These lesions have a particularly high rate of recurrence, even with prolonged rehabilitation times. Muscle fibers of the semimembranosus can be classified according to their origin into three sections, with fibers arising from the medial and lateral parts of the proximal tendon and fibers arising from the distal myoaponeurotic junction having the worst prognosis [50]. The most frequent tears of the adductor longus are tears of the proximal tendon (including tendon avulsions) or intramuscular midsubstance tears [51]. Distal adductor longus tendon tears are exceedingly rare. On the other hand, the adductor magnus has fibers very close to the hamstring/ischial muscles; therefore, proximal lesions are difficult to identify clinically. The connective tissue that covers the deep surface of the medial gastrocnemius distally blends with the Achilles tendon, resulting in a significant change in caliber, which would form a weak point [52].

Although magnetic resonance imaging (MRI) is commonly used for initial diagnosis, its role in predicting reinjury risk and determining return-to-play readiness remains inconclusive [53]. Based on the current evidence, there is no strong evidence for any MRI finding at baseline and/or RTP in predicting muscle reinjury risk [54], so relying on MRI results for RTP is uncommon [55] (Figure 3). Functional healing does not correspond to the negativization of MRI, thus leading to the hypothesis that functional healing precedes imaging [56,57]. However, a recent retrospective study [53] found that connective tissue gap, intermuscular edema, and callus gap were related to a higher risk of muscle reinjury (OR 29.58, *p* = 0.001), while van der Horst et al. [58] highlighted how an increased MRI STIR signal intensity was inversely related to the risk of reinjuries. These are promising results that radiological imaging findings could become more helpful in the RTP process in the future. Some functional magnetic resonance imaging techniques, such as T2 mapping and diffusion tensor imaging (DTI), have been proposed to better assess return to play, but measurements of T2 relaxation time and diffusion are not as good as a radiologist’s visual report at predicting return-to-play time after acute muscle tear [59].

## 7. The Potential Role of Artificial Intelligence

In recent years, the intersection of sports medicine and technology has witnessed a transformative shift, largely propelled by advancements in artificial intelligence (AI), machine learning (ML), and deep learning (DL) [60]. These sophisticated computational techniques have begun to play an instrumental role in muscle injury prevention, diagnosis, and management [61].

The integration of AI in sports medicine, particularly in muscle injury prevention, offers a transformative approach to understanding and mitigating injury risks. AI systems, with their capacity to process and analyze vast datasets, can provide insights that might be elusive to traditional methods. By examining variables like an athlete’s training loads, biomechanics, and injury history, AI can predict potential reinjury risks with heightened accuracy. This predictive capability can be instrumental in tailoring training regimens, ensuring athletes strike the right balance between training intensity and recovery, thereby minimizing the risk of reinjuries. Furthermore, AI can shed light on nuanced risk factors that might be overlooked in conventional athlete screenings. Lu et al. [62] utilized the extreme gradient boosting ML algorithm to predict muscle strain risks in NBA players; their findings—though valuable—primarily reiterated known risk factors such as a history of lower extremity injuries and recent concussions. However, the true potential of AI lies in its ability to identify less obvious, interconnected risk factors by analyzing vast and diverse datasets. Such insights can revolutionize our understanding of injury mechanisms and pave the way for more effective prevention strategies. The same model was used by Ayala et al. [63] to identify professional soccer players at risk of hamstring injuries during preseason screenings. Identifying such risk factors could be crucial to prevent injury relapses.

ML, a subset of AI, involves algorithms that improve automatically through experience. In the context of muscle injury diagnosis, ML can be instrumental. By analyzing medical images, such as MRIs or X-rays, ML algorithms can detect subtle changes or patterns that might be indicative of a predisposition to reinjury. These algorithms can be trained on vast datasets of medical images, learning to identify the minutiae that might escape the human eye. In their recent work conducted on football players with the use of ML approaches, Valle et al. [64] showed how the most important factors to determine the return to play after a hamstring injury were if the injury was at the free tendon of the biceps femoris long head or if it was a grade 3^r^ injury, using their classification.

DL, a further subset of ML, employs neural networks with many layers (hence “deep”) to analyze various factors of data. In muscle injury management, DL can be particularly useful in postinjury rehabilitation [65]. Wearable sensors can capture data on an athlete’s movement dynamics, which DL models can then analyze to assess the effectiveness of rehabilitation exercises [66]. If an athlete’s movement deviates from the optimal pattern, the DL model can flag this, allowing physiotherapists to adjust the rehabilitation protocol accordingly. This real-time feedback loop can ensure that athletes regain optimal movement patterns, reducing the risk of reinjury. For example, Skazalski et al. [67] showed that the commercially available Vert device was able to track an athlete’s progress to estimate the likelihood of injury among volleyball players during training and competition. In that sense, movement analysis has shown interesting results. In detail, surface electromyography coupled with inertial measurement units or kinematic analysis could allow a deeper analysis of neuromuscular behavior by detecting early kinematic alteration, which represents a risk factor for injury. In particular, neuromuscular tests are increasingly used during the rehabilitation plan to verify the progress of patients for a safer RTP [41]. Proper rehabilitation and the correct timing for RTP play an important role in avoiding future reinjuries.

Furthermore, AI-driven predictive analytics can play a pivotal role in personalized medicine. By analyzing an individual’s unique biomechanics, genetics, and injury history, AI systems have the potential to recommend personalized training and rehabilitation programs [68]. This bespoke approach ensures that interventions are tailored to an individual’s specific needs, thereby minimizing reinjury risks.

## 8. Prevention

Prevention and exercise programs are essentially correct strength training for a muscle group. Eccentric exercises have the strongest evidence as a secondary prevention strategy [69], as in the case of the Copenaghen program for the adductors [70] or the Nordic hamstring for the hamstrings [71].

Running, especially acceleration and reaching peak speed, also plays a preventive role in hamstring reinjuries [72]. Proper training techniques, optimal loading, and monitoring of training loads are crucial components of injury prevention [73]. The acute–chronic workload ratio data of the player should always be maintained between 0.8 and 1.3, preferably with GPS monitoring [73]. Indeed, comparing GPS data allows us to understand if the progression is followed or if there is some kind of unconscious compensation or neuromuscular adaptation other than the athletic health status of the player.

Addressing individual factors such as muscle strength, flexibility, and psychological readiness can further reduce the risk of muscle reinjury. In addition, psychological management of the injury is often beneficial in preventing new episodes [74].

Moreover, a recent study by de Sire et al. [75] reported the positive effect of introducing a neuromuscular warm-up consisting of structured injury prevention exercises. This could have an immediate effect in improving the preactivation time of the knee stabilizer muscles, namely, the rectus femoris, vastus medialis, and medial and lateral hamstrings, thus improving the risk of ACL injuries.

Therefore, correct prevention of muscle reinjury is a multiparameter task (Figure 4).

## 9. Limitations

While comprehensive and insightful, this review comes with its set of limitations. First and foremost, the nature of this review differs from systematic reviews or meta-analyses. While the latter follows a strict protocol and criteria for study inclusion, ensuring a balanced and exhaustive representation of the literature, our review might inadvertently introduce a selection bias based on the authors’ discretion. This could potentially lead to the omission of some relevant studies. Additionally, the depth of analysis in our review might not match the granularity often seen in systematic reviews and meta-analyses. Such reviews delve deeper into individual studies, rigorously assessing their quality and risk of bias. In contrast, our broader overview might occasionally miss out on capturing individual studies’ nuanced findings or interpretations. Given the dynamic nature of research, some recent studies might not have been included due to the time frame of our search. Lastly, while we have made every effort to accurately represent the studies we have included, there is always a risk of misinterpretation or oversimplification, especially when translating complex research findings into more digestible content.

## 10. Practical Implications

The findings and insights from this review hold significant practical and clinical value. By shedding light on the risk factors and epidemiology of muscle reinjuries, clinicians and sports professionals are better equipped to manage and rehabilitate athletes, ensuring a safer and more effective return to play. This knowledge also paves the way for the development of targeted training and conditioning programs. By addressing the identified risk factors head-on, these programs can potentially prevent the onset of the initial injury, safeguarding athletes’ long-term health and performance. The highlighted potential of AI in this review is particularly promising. As technology continues to advance, integrating AI tools into clinical practice could revolutionize injury prediction and management. These tools, backed by vast datasets, can offer data-driven insights, enabling practitioners to make more informed, proactive decisions. Beyond its direct clinical implications, this review serves as a valuable educational resource. Athletes, coaches, and other stakeholders can benefit from a deeper understanding of muscle reinjuries, their consequences, and the importance of proper management.

## 11. Conclusions

In conclusion, lower limb muscle reinjuries continue to pose significant challenges for athletes and medical professionals, especially in the football field. Advances in research and a multidisciplinary approach combining prevention strategies, accurate diagnosis, and individualized rehabilitation plans play a key role in reducing the incidence of muscle reinjuries. The potential of AI, as highlighted, could be a transformative tool in predicting and mitigating the risk of reinjuries. Further studies should continue to improve our understanding of risk factors, refine clinical management strategies, and promote preventive measures, which will undoubtedly contribute to better outcomes for athletes and the field of sports medicine as a whole.

## Figures and Tables

**Figure 1 jfmk-08-00155-f001:**
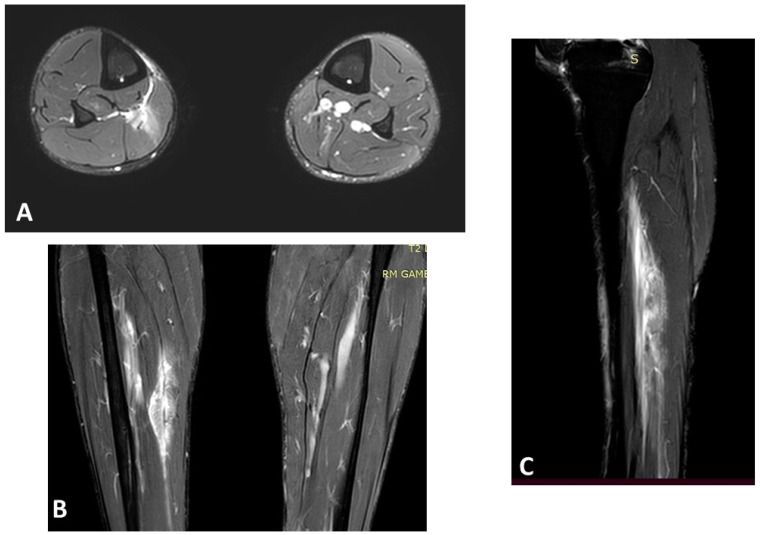
Magnetic resonance imaging (MRI) of a right soleus injury (3B lesion) in a 45-year-old professional football player: (**A**) axial T2 spectral attenuated inversion recovery (SPAIR) image; (**B**) coronal T2 SPAIR image; (**C**) sagittal T2 SPAIR image.

**Figure 2 jfmk-08-00155-f002:**
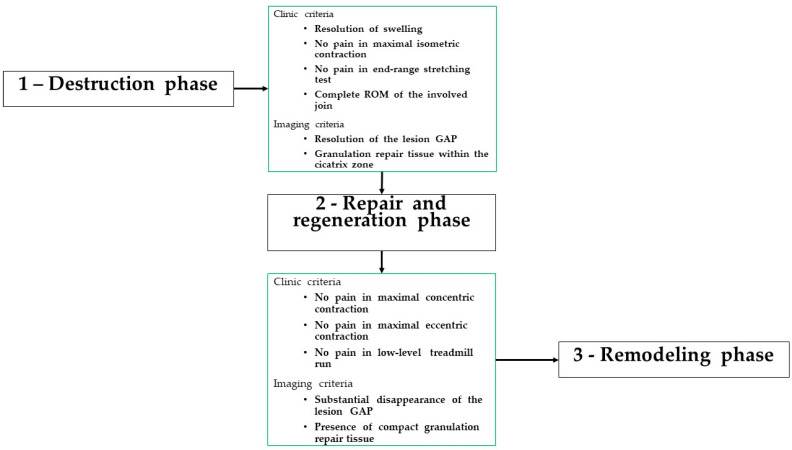
Management of a muscle reinjury [33]: ROM, range of motion.

**Figure 3 jfmk-08-00155-f003:**
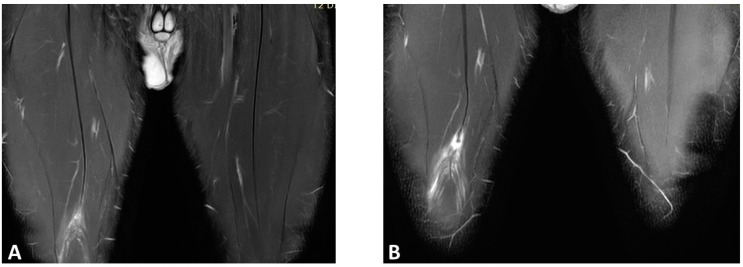
Coronal T2 spare magnetic resonance imaging (MRI) of a right rectus femoris injury in a professional football player: (**A**) first episode (3A lesion); (**B**) reinjury after 30 days. Note that the recurring lesion is more serious, as it involves a rupture of the central septum.

**Figure 4 jfmk-08-00155-f004:**
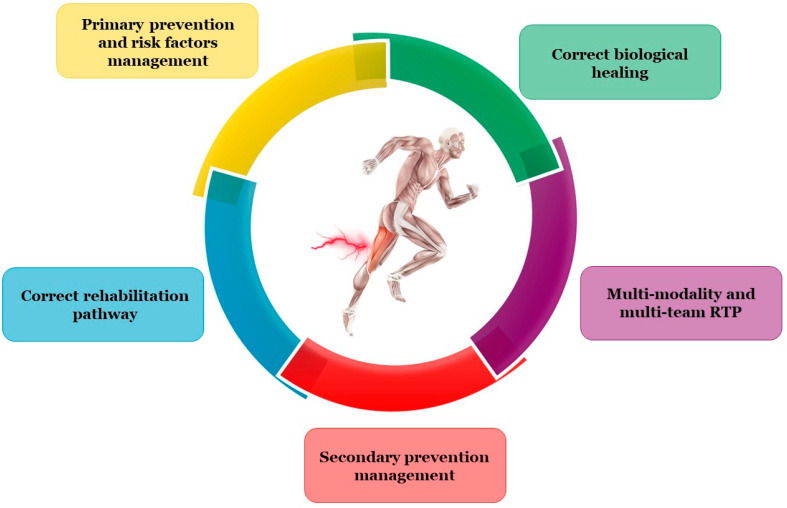
How to prevent muscle reinjury: RTP, return to play.

**Table 1 jfmk-08-00155-t001:** Details regarding absence days from the sport for more common muscle injuries in professional soccer [10].

Injury	Mean Absence Days from Sport (95% CI)
Quadriceps muscle injury (structural)	23.7 (20.2 to 27.2)
Hamstring muscle injury (structural)	21.5 (18.9 to 24.1)
Calf muscle injury (structural)	20.8 (17.0 to 24.5)
Hamstring muscle injury (functional)	9.2 (7.1 to 11.3)
Calf muscle injury (functional)	7.3 (4.1 to 10.6)
Quadriceps muscle injury (functional)	6.4 (4.3 to 8.4)

**Table 2 jfmk-08-00155-t002:** Italian Society of MUscles, Ligaments and Tendons (ISMULT) recommendations for muscle reinjury.

High-grade recommendations	Higher risk of reinjury if history of previous muscle injuries
	Always manage modifiable risk factors and complete the correct rehabilitation process
	Eccentric exercise should be a cornerstone of the rehabilitation process
Low-grade recommendations	Higher risk of reinjury for high-grade muscle injuries
	Myotendinous injuries are the more at-risk lesions
	Low risk of reinjury in professional athletes
	More than half of hamstring reinjuries occur within 4 weeks of RTP

**Table 3 jfmk-08-00155-t003:** Ultrasound classification of muscle lesions [44].

Grade	Definition	Ultrasound Characteristic
I	Minimal elongations with less than 5% of muscle involved	These lesions can be quite long in the muscle axis being usually very small on cross-sectional diameter
II	Partial muscle ruptures	Lesions that involve from 5 to 50% of the muscle volume or cross-sectional diameter. Ultrasound demonstrates a hypo or even anechoic gap within the muscle fibers. Gentle pressure applied with the transducer will demonstrate torn muscle fragments floating in a serohematic fluid
III	Complete muscle tears with complete retraction	The muscle belly forms a real mass, and a gap can be palpated between the retracted ends of the muscle

**Table 4 jfmk-08-00155-t004:** MRI assessment of muscle injuries [42].

Clinical information	Data on trauma Mechanism of injury Symptoms Sports discipline History of prior injuries in the same region		
Evaluation of the MRI	Anatomical assessment of T1-weighted sequences (axial and coronal)	Individual muscular anatomy Anatomical variants Residual changes from previous lesions (scarring, atrophy) Vascular structure	
	Assessment of lesions on T1- and T2-fluid-sensitive sequences (acute lesions on T2-weighted sequences and previous lesions on both T1- and T2-weighted sequences)	Location of the lesion	Proximal, middle, and distal
		Anatomical structures involved	Aponeurosis, fascia, tendon, and fibers
		Pattern of edema and/or scar	
Categorization of MRI lesions based on clinical and imaging criteria	Munich Consensus Statement [47] or British Athletics Muscle Injury Classification [48] or FC Barcelona—Aspetar–Duke classification [1]		

MRI, magnetic resonance imaging.

**Table 5 jfmk-08-00155-t005:** Representative MRI patterns of frequent muscle injuries [42].

Muscle	Injury Pattern	Approximately Expected RTP Time
Rectus femoris	Gap in the central septum	6–7 weeks
	Gap in the anterior aponeurosis	6–7 weeks
	Discontinuity of the anterior fascia	2–3 weeks
Soleus	Rupture of posterior aponeurosis	3–4 weeks
	Rupture of the central septum	5–6 weeks
	Rupture of the medial fascicle	5–6 weeks
	Rupture of the lateral fascicle	3–4 weeks
Semimembranosus	Myotendinous injuries	3–4 weeks
Biceps femoris	Deep zipper	3–4 weeks

## Data Availability

The data that support the findings of this study are available from the corresponding author upon reasonable request.

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
