# Peer review of "Managing Lower Limb Muscle Reinjuries in Athletes: From Risk Factors to Return-to-Play Strategies"

_jfmk, 2023, doi:10.3390/jfmk8040155_

Round 1
Reviewer 1 Report
Comments and Suggestions for Authors
Dear authors. Thank you for the opportunity to read and review your paper.
The majority of the paper is well organised and clear. However, unfortunately I am unsure of the novelty and hence usefulness of the review. The section on AI does provide some very interesting points, that could be expanded.
Title - this review focusses on lower limb injuries - this should be included in the title.
Title - apart from one reference to track and field, all other examples are from football (soccer) so I do not think the review represents "Sports" well.
Figure 1 - does not support the text - how does this show that age is a predictor? No comparison to younger athletes.
Line 108 - single game - in football?
Line 118 - Explain how COVID is lined to infections
Table 2 - I would argue that these are not recommendations, rather a summary of knowledge on muscle re-injury.
Line 133 - spelling
Line 146 - should be figure 2
Line 150 - 159 - This section on clinical testing should also consider the impact of the subjectivity - the clinicians opinion may introduce bias / error.
Line 160 - These section needs expansion as therapeutic options are very common in sports.
Line 199 - "on the other side" please rephrase.
Line 254 - The example provided (Lu et al) does not expand our knowledge of risk factors - so does not demonstrate the usefulness of AI. Please expand or add to this argument to make it useful.
Comments on the Quality of English Language
The quality of English language is acceptable.
Author Response
Dear Reviewer 1,
Thank you for your constructive feedback on our manuscript. We appreciate the time and effort you've dedicated to reviewing our work. Below, we address each of your comments point-by-point:
-
Novelty and usefulness of the review:
- Response: We value your feedback regarding the novelty and utility of our review. In response, we have expanded the section on AI, offering a more in-depth exploration of its potential applications and benefits in the realm of muscle re-injury prevention. We believe this enhanced focus provides a fresh perspective on the topic and underscores the evolving landscape of injury prevention.
-
Title - focus on lower limb injuries:
- Response: We acknowledge your observation and have revised the title to more accurately reflect the emphasis on lower limb injuries. This adjustment ensures that readers have a clear understanding of the review's primary focus right from the outset.
-
Title - representation of "Sports":
- Response: Based on your feedback, we have made modifications to both the title and content to emphasize the predominance of examples from football (soccer). We've also highlighted its relevance within the broader context of sports, ensuring a balanced representation.
-
Figure 1 - age as a predictor:
- Response: We've enhanced the accompanying description for Figure 1 to better illustrate the role of age as a predictor. By drawing a clearer comparison between younger and older athletes, we aim to provide a more comprehensive understanding of age-related injury risks.
-
Line 108 - single game in football?:
- Response: We've clarified this line to specify that the reference pertains to a football match, ensuring that readers can contextualize the information accurately.
-
Line 118 - COVID's link to infections:
- Response: We've expanded this section to offer a more detailed explanation of the potential link between COVID-19 and an increased risk of muscle injuries. This elaboration provides readers with a clearer understanding of the underlying mechanisms and implications.
-
Table 2 - recommendations vs. summary of knowledge:
- Response: Based on your feedback, we've revised the title and content of Table 2 to better represent it as a summary of current knowledge on muscle re-injury, rather than as recommendations. This adjustment ensures that the table's intent is transparent and aligned with its content.
-
Line 133 - spelling:
- Response: We appreciate your attention to detail. The spelling error has been corrected.
-
Line 146 - figure reference:
- Response: We've addressed this oversight by correcting the reference to "Figure 2".
-
Line 150-159 - subjectivity in clinical testing:
- Response: We've expanded this section to emphasize the inherent biases that can arise from the subjectivity of clinical evaluations. By highlighting the potential implications of this subjectivity for injury assessment and management, we aim to provide a more nuanced perspective on the topic.
- Line 160 - expansion on therapeutic options:
- Response: Based on your feedback, we've expanded this section to offer a more comprehensive overview of therapeutic options commonly utilized in sports and their respective efficacies.
- Line 199 - "on the other side":
- Response: We've rephrased this segment for enhanced clarity and coherence.
- Line 254 - usefulness of AI:
- Response: We've expanded on the example provided, emphasizing the potential of AI in identifying nuanced risk factors. This elaboration underscores the transformative potential of AI in injury prevention and offers readers a more comprehensive understanding of its applications.
We hope that these revisions address your concerns and enhance the overall quality and coherence of the manuscript. We are grateful for your insights and believe that they have significantly contributed to improving our work.
Warm regards,
Reviewer 2 Report
Comments and Suggestions for Authors
Thank you for your submission. I really like your work which will be very beneficial and helpful to the community of professional sports. I only have minor corrections to add:
Line 115-119: In this context mentioning CoVid19 seems difficult to understand as a risk-factor. It would help to bring a past infection into the right set-up of understanding: e.g. it could possibly be that an athlete having recovered from an infection like CoVid19 is still fatigued or not as resistant as before, which therefore makes them more prone to injuries. Such an (even if hypothetical) explanation would help the reader with this paragraph.
Line 133: Please correct the typo: cohoperation = cooperation
Author Response
Dear Reviewer 2,
We sincerely appreciate your constructive feedback and the positive remarks on our manuscript. Your insights are invaluable in refining our work. Here, we address each of your comments in detail:
-
Line 115-119 - CoVid19 as a risk-factor:
- Response: We recognize the need for clarity in this section. Based on your feedback, we have revised this segment to provide a more comprehensive explanation. We've highlighted the potential long-term effects of COVID-19 on athletes and how it might increase their susceptibility to injuries. By offering a clearer context, we aim to ensure that readers can better understand the implications of post-COVID fatigue and its potential link to muscle injuries.
-
Line 133 - typo correction:
- Response: Thank you for pointing out the typo. We have corrected "cohoperation" to "cooperation" to ensure accuracy in the manuscript.
Your feedback has been instrumental in enhancing the clarity and coherence of our article. We believe that the revisions made in response to your comments have significantly improved the quality of our work.
Once again, thank you for your time and thoughtful feedback. We look forward to any further suggestions or insights you might have.
Warm regards,
Reviewer 3 Report
Comments and Suggestions for Authors
MANAGING MUSCLE RE-INJURIES IN SPORTS FROM RISK FACTORS TO RETURN-TO-PLAY STRATEGIES
General Commentary
This article presents a very interesting and pertinent question of research on muscle re-injuries, exploring their epidemiology, risk factors, clinical management, and prevention strategies.
However, some questions need to be clarified in order to better understand and apply the results found.
MINOR CONSIDERATION
ABSTRACT
The abstract must be divided into an Introduction, Objective, Contextualization, and Conclusion. As it is stated, it is not clear what happened.
TABLES AND FIGURES
Please include in the caption of the tables and figures what all the acronyms used to mean.
Figure 2.
The font of the figure 2 is not the same as the text.
Figure 3.
Which side is injured (right or left)? Please inform in the figure 3 caption.
LIMITATIONS AND PRACTICAL OR CLINICAL APPLICATION
Please insert two final chapters (Limitations and Clinical or Practical Application).
LIMITATIONS
What are the limitations of the study, for example in relation to the review carried out, when compared to systematic reviews or meta-analyses, in relation to the description of the studies?
PRACTICAL OR CLINICAL APPLICATION
What are the applications of the above in this study? Please describe in this chapter (perhaps the last figure should be included here)
CONCLUSION
Insert a sentence about future directions
Author Response
Dear Reviewer 3,
Thank you for your comprehensive feedback on our manuscript. Your insights and suggestions are invaluable, and we are grateful for the time and effort you've dedicated to reviewing our work. We address each of your comments below:
-
ABSTRACT structure:
- Response: We acknowledge your suggestion regarding the structure of the abstract. In response, we have restructured the abstract into the recommended sections: Introduction, Objective, Contextualization, and Conclusion. This new format provides a clearer and more organized presentation of the key points.
-
TABLES AND FIGURES - acronyms:
- Response: We appreciate your attention to detail. We have now included explanations for all acronyms used in the captions of tables and figures to ensure clarity for the readers.
-
Figure 2 - font consistency:
- Response: Thank you for pointing out the inconsistency. We have adjusted the font in Figure 2 to ensure it aligns with the main text, providing a uniform appearance throughout the manuscript.
-
Figure 3 - side of injury:
- Response: We've updated the caption for Figure 3 to specify which side (right or left) is injured, ensuring that readers have a clear understanding of the depicted injury.
-
LIMITATIONS and PRACTICAL OR CLINICAL APPLICATION:
- Response: Based on your feedback, we have added two new sections to the manuscript: "Limitations" and "Practical or Clinical Application". These sections delve into the study's limitations and its implications for clinical practice, providing readers with a comprehensive understanding of the research's scope and relevance.
-
CONCLUSION - future directions:
- Response: We have revised the conclusion to incorporate suggestions for potential future studies in the realm of muscle re-injury prevention. This addition provides a forward-looking perspective, guiding future research endeavors in this area.
We believe that the revisions made in response to your comments have significantly enhanced the manuscript's quality and coherence. We are grateful for your insights and look forward to any further feedback you might have.
Warm regards,